# The impact of racially-targeted food marketing and attentional biases on consumption in Black adolescent females with and without obesity: Pilot data from the Black Adolescent & Entertainment (BAE) study

Omni Cassidy[1,2]*, Marian Tanofsky-Kraff[1], Andrew J. Waters[1], Lisa M. Shank[1¤a¤b], Abigail Pine[1¤c], Mary Quattlebaum[1¤d], Patrick H. DeLeon[1,3], Marie Bragg[2,4], Tracy Sbrocco[1]

1 Department of Medical and Clinical Psychology, Uniformed Services University of the Health Sciences, Bethesda, MD, United States of America, 2 Department of Population Health, NYU Langone Health/ Grossman School of Medicine, New York, New York, United States of America, 3 Daniel K. Inouyé Graduate School of Nursing, Uniformed Services University of the Health Sciences, Bethesda, Maryland, United States of America, 4 College of Global Public Health, New York University, New York, New York, United States of America

¤a Current address: Department of Medicine, Military Cardiovascular Outcomes Research (MiCOR) Program, Uniformed Services University of the Health Sciences, Bethesda, Maryland, United States of America
¤b Current address: The Metis Foundation, San Antonio, Texas, United States of America
¤c Current address: Department of Psychology and Human Development, Vanderbilt University, Nashville, Tennessee, United States of America
¤d Current address: Department of Psychology, University of South Carolina, Barnwell College, Columbia, South Carolina, United States of America
* omni.cassidy@nyulangone.org

## Abstract

Unhealthy food advertisements ("advertisements" hereafter referred to as "ads") are linked to poor diet and obesity, and food companies disproportionally target Black youth. Little is known about the mechanisms whereby food ads influence diet. One possibility may be racially-targeted ads that appeal to Black youth. Those with food-related attentional biases may be especially vulnerable. The objective of this project was to assess the feasibility and initial effects of a pilot study testing the influence of racially-targeted food ads and food-related attentional biases on eating behaviors among a sample of Black adolescent females. Feasibility of recruitment, retention, and procedures were examined. Participants ($N = 41$, 12-17y) were randomized to view a television episode clip of the *Big Bang Theory* embedded with either four 30-second racially-targeted food ads or neutral ads. A computer dot probe task assessed food-related attentional biases. The primary outcome was caloric consumption from a laboratory test meal. Interactions based on weight and ethnic identity were also examined. Analyses of variance and regressions were used to assess main and interaction effects. Exposure to racially-targeted food ads (versus neutral ads) did not affect energy consumption ($p > .99$). Although not statistically significant, adolescents with obesity consumed nearly 240 kcal more than non-overweight adolescents ($p = 0.10$). There were no

**Data Availability Statement:** All relevant data are within the paper and its Supporting Information files.

**Funding:** This research was funded by the National Institute of Minority Health and Health Disparities, grant number 1F31MD010675-01 (https://www.nimhd.nih.gov/); and Uniformed Services University of the Health Sciences, grant number TO7238251 (https://www.usuhs.edu/; both to O. C.). The funders had no role in study design, data collection and analysis, decision to publish, or preparation of the manuscript.

**Competing interests:** The authors have declared that no competing interests exist.

significant preliminary effects related to food-related attentional biases or ethnic identity ($p$s = 0.22–0.79). Despite a non-significant interaction, these data provide preliminary support that adolescents with obesity may be particularly vulnerable to racially-targeted food ads. An adequately powered trial is necessary to further elucidate the associations among racially-targeted food ads among Black adolescent girls with obesity.

## Introduction

Unhealthy food advertisements ("advertisements" hereafter referred to as "ads") are one environmental factor that contributes to excessive weight gain and poor diet in Black youth [1]. Food companies spend nearly $11 billion each year on television (TV) food/beverage ads and $333 million on ads aimed at Black youth [1]. The highest promoters of food/beverages to Black youth include Yum! Brands (Taco Bell, KFC, Pizza Hut), 3G Capital (Burger King), McDonald's, and Popeye's [2,3]. As such, Black youth are exposed to 86–119% more food/beverage TV ads than their White peers [1], and the majority of these ads are for products high in fat, sugar, and salt [4]. Ads placed in predominantly Black neighborhoods or featuring Black actors are also less likely to promote healthy products compared to ads targeting non-Black audiences [5]. Increased exposure to unhealthy food and beverage ads influences attitudes, preferences, and consumption [6], contributing to poor diet and subsequently high rates of chronic illnesses in Black communities [7]. It is critical to understand the potential influence of these ads on eating behaviors to prevent poor diet and determine appropriate intervention targets.

Food ads targeting Black youth (hereafter referred to as "racially-targeted food ads") are often embedded with cultural features intended to appeal to Black individuals. These ads may feature Black actors (e.g., spokespersons for Popeyes), celebrities (e.g., Beyoncé, LeBron James), music, or other activities (e.g., basketball) perceived to align with Black cultural preferences or values [5,8–11]. Ads may, for example, feature young Black people who appear healthy, happy, and engaged in fun social activities (e.g., recording a music album) while eating fried chicken, hamburgers, or other fast foods. These "consumption symbols" are designed to highlight the psychosocial benefit of products—signaling to young Black people that consuming the advertised product will improve social connection, mood, or attractiveness [12,13]. These messages may be particularly salient for adolescents whose higher-order cognitive functioning is still developing, and Black adolescents who may utilize the media more often to understand ethnic membership and support ethnic identity development [14]. Although research has shown a negative effect of unhealthy foods ads in children, there is a paucity of research among adolescents [15]. Further, the underlying mechanisms potentially driving effects are not well understood.

Food-related attentional bias (ABs) is one underlying mechanism potentially driving the effect of food ads on attitudes, preferences, and consumption. Food-related ABs refer to distinct, yet related processes of selectively attending or over-attending to food cues in ones' environment [16]. Although data are mixed [17–19], food-related ABs has been associated with overconsumption [20,21], higher weight status [22,23], and weight gain [24]. According to the incentive-sensitization theory [25], repeated exposure to food cues can lead to a heightened reward response in vulnerable individuals. This exacerbated response may subsequently increase salience and establish strong cravings for the food item. As the salience for the food item increases, seeking out and consuming the food to diminish the craving becomes critical, overriding a person's homeostatic feeding drive. With increased food availability in many food

environments, such drives may lead to overeating [26,27]. While most individuals exhibit food-related ABs towards food-related cues when hungry [28–30], individuals repeatedly exposed to food cues [25] and those with obesity [29] may be particularly vulnerable. Not only are Black youth exposed to more unhealthy food ads compared to other groups [31], they often reside in communities with a disproportionately high density of food cues [32,33]. Among Black youth with obesity [34,35], external food cues may become highly salient, leading to food-related ABs and overeating [34,36,37]. Black youth who are exposed to racially-targeted food ads and have food-related ABs may be particularly vulnerable to overconsumption, although this relationship has not been explored in Black adolescent females with and without obesity.

Studies elucidating the potential link between food ads and food-related ABs on energy consumption have primarily examined adults and are based on self-report [21,38]. One experimental study, however, by Folkvord and colleagues [37] examined the interaction of unhealthy food ads and food-related ABs in children (grades 2 through 4). Children were randomized to an advergame (i.e., online video game promoting particular products) embedded with either a food ad or a non-food ad. Following the advergame, children were invited to consume snack foods *ad libitum*. Using eye tracking to measure food-related ABs, scientists measured three components of visual AB. Children who played the advergame with food ads consumed significantly more snack foods compared to children in the control condition. Among children who played the advergame, gaze duration (one type of food-related AB) was significantly associated with more snack intake, even after controlling for age, sex, and hunger. Data suggest that food-related ABs may moderate the effect of food ads on eating, but this association is not well understood.

To address the research gaps, the primary aims of this pilot study were: 1) to assess feasibility of the recruitment and study protocol to assess the intersection of racially-targeted foods ads and food-related ABs, 2) examine initial effects of racially-targeted food ads on energy intake, and 3) examine initial effects of food-related ABs on energy intake in a sample of Black adolescent females. Our primary hypotheses were that: 1) the study would be feasible, 2) exposure to racially-targeted food ads would be associated with higher energy intake, 3) exhibiting high food-related ABs would be associated with higher energy intake, and 4) there would be an interaction such that exposure to racially-targeted food ads and high food-related ABs would lead to the highest energy intake.

## Materials and methods

### Recruitment

Participants were recruited from Washington, DC, and the surrounding metropolitan area June 2017-June 2018. The study purpose was concealed in accordance with prior research [39]. Potential participants believed they were recruited for a study on how relationships are depicted on TV. Individuals were eligible if they were female, 12–17 years, Black/African-American, $\geq 5^{th}$ body mass index (BMI; kg/m$^2$) percentile for age/sex, and English-speaking. Individuals were excluded if they reported a medical/psychiatric illness; taking medications that affect appetite, mood, or weight; disliking $\geq 40\%$ of test meal foods; or food allergies/restrictions that could not be accommodated. The sample included 41 Black females 12–17 years (M, SD = 14.93, 1.67).

### Procedure

A telephone pre-screen was conducted and eligible adolescents were scheduled for a visit. Adolescents were asked to fast and not exercise two hours before the visit to prepare for body

measurements. To conceal the study purpose, potential participants were told that body weight would be assessed because those of higher weights may be more sensitive to relationship interactions. Parents/guardians and adolescents signed consents and assents, respectively. To confirm eligibility, fasting body weight (kg) and height (cm) in triplicate were measured. Participants consumed a granola bar (100 kcal) to ensure they were not extremely hungry before the food-related ABs computer dot probe task ("AB task"). On a hunger visual analogue scale (0 = "not at all hungry" to 100 = "extremely hungry"), participants did not report being extremely hungry before the AB task (M, SD = 38.72, 27.79).

Adolescents completed a 15-minute food-related ABs computer task examining the speed their attention was drawn toward food images versus non-food images. Participants then viewed a 14-minute clip of *The Big Bang Theory* [40]. References to food, eating, or restaurants were removed. The episode was embedded with: 1) four 30-second racially-targeted food ads (McDonald's, Burger King, Popeye's, Kentucky Fried Chicken) with culturally-congruent features (e.g., Black actors) or 2) four 30-second neutral ads (Geico, Esurance, Ford, Honda) without culturally-congruent features. Ads were selected from Adscope, a database that provides access to ad content [41]. The Adscope database categorizes commercials using the following criteria: name, date aired, platform (e.g., TV, radio), attributes (demographic specifications), markets (specific geographic regions), language, and length. Eight separate searches for food and neutral commercials was conducted using the following search terms: McDonald's, Burger King, KFC, Popeye's, Ford, Honda, Esurance, and Geico. Searches also included the following search terms (specific category included in parentheses): three years prior to search year in 2016 (date aired), TV only (media), ethnic targeted/African American (attributes), Baltimore, MD/Washington, DC/Hagerstown (markets), English (language), and 30 seconds (length). Commercials were excluded if they did not advertise the preferred items (e.g., McDonald advertising only chicken products), promoted activities that were unlikely to appeal to adolescents (e.g., toys for young children), references to sports or celebrities, off-season (e.g., references to winter), or promoted a corporation (e.g., Bank of America). For neutral commercials, ads that referenced food, eating, or restaurants were excluded. A total of 15 McDonald's, four Burger King, 10 Popeye's, eight KFC, nine Geico, 10 Ford, eight Esurance, and seven Honda commercials remained. The remaining commercials were reviewed and matched based upon the following criteria: number of people, estimated screen time for persons and products, perceived age range and gender of persons on screen, and emotional affect. Increased exposure to food products could influence eating behaviors, and episodes can activate different emotions, such as happiness or sadness. The estimated time the product was on screen and emotional affect were, therefore, determined to be the most critical criteria. Various combinations of four food and four neutral commercials were compared until the optimal matching on the aforementioned categories was achieved.

Following the marketing exposure, adolescents were provided 30 minutes to eat *ad libitum* from a previously used multi-item test meal. Participants were debriefed at the end of the visit. Adolescents received a $40 gift card for participating in the study. The episode was edited using Apple Final Cut Pro X (Version 10.3.4; Cupertino, CA).

## Ethics statement

The study was approved by the Human Research Protection Program at the Uniformed Services University and underwent an ethical review to ensure proper protections for minors. Study team members verbally described the procedures, risks, and benefits of the study with parents/guardians and adolescents. A parent/guardian provided written informed consent and adolescents provided written informed assent. Study team members emphasized to parents/

guardians and adolescents that participation was completely voluntary and they could withdraw from the study without penalty or effect. We emphasized that adolescents had the right to decline participation or withdraw from the study at any point regardless of the parents'/guardians' interest in their participation.

## Measures

Age and parents'/guardians' occupation were collected via self-report. Researchers used occupations to estimate household income based on 2018 national wage estimates from the U.S. Bureau of Labor Statistics [42]. Body weight (two-hour fasting) was measured in triplicate to the nearest 0.1 kg using the Tanita BF 350 Body Composition Analyzer (Tanita Corporation of America, Inc., Arlington Heights, IL). Height (two-hour fasting) was measured in triplicate to the nearest 0.1 cm using a stadiometer. Participants completed rating scales for hunger, fullness, and food craving on a visual analogue scale ranging from 0 ("Not at all") to 100 ("Extremely") [43,44]. Hunger ratings were examined pre- and post-AB task and pre- and post-meal. Only pre-AB task hunger ratings are reported. Participants completed a self-report dietary recall of the foods and drinks they consumed since awakening. Caloric information was derived from manufacturers.

To minimize the confounding effect of food preference on consumption from the test meal, only participants who preferred the majority of the foods offered during the test meal were included in the study. To assess food preferences, test meal food items were embedded within a larger, general food preference questionnaire that has been used to identify food preferences in youth [45]. Participants rated foods on a visual analogue scale ranging from 1 ("I hate this food") to 10 ("I love this food"). Participants who did not provide a rating of 6 or more for at least 60% of the food items were excluded. The questionnaire may have primed potential participants towards food, but was necessary to ensure all participants would accept the majority of foods provided during the test meal.

To examine ABs, participants completed a visual dot probe task, a validated behavioral measure of food cue incentive salience in overweight samples [29,46–48]. The AB task randomly presented 60 image pairs of high palatable foods versus nonfoods ("highpal-nonfood"), low palatable foods versus nonfoods ("lowpal-nonfood"), and high palatable foods versus low palatable foods ("highpal-lowpal;" $n$ = 180). Trials consisted of a mixture of incongruent trials (when the probe replaced the neutral image) and congruent trials (when the probe replaced the food image of highest palatability). Pairs were presented followed by an arrow that randomly replaced the least or most palatable image, then a probe prompting a response. The spatial location of images and probes were counter-balanced. Internal split-half reliability was excellent for mean reaction times ($r$s = 0.84–0.92), but poor for bias scores ($r$s = -0.30–0.40). Only data from highpal-nonfood and highpal-lowpal trials are reported.

Adolescents were provided 30 minutes to eat *ad libitum* from a previously used multi-item test meal including chicken nuggets, white bread, Oreos ©, M&Ms ©, barbecue sauce, peanut butter, jelly, water, apple juice, and 2% fat-free milk (~4,400 kcal) [49]. Consumption was calculated by weighing each item pre- and post-meal. Nutrition information was derived from manufacturers. Meals occurred at approximately 10:30am, 12:30pm, 2:30pm, or 4:30pm.

Food ads embed cultural relevance by utilizing cultural food preferences [8,9]. Ethnic identity was used as a proxy for identifying individuals who may be more susceptible to such practices. The Cross Racial Identity Scale (CRIS) [50] is a 40-item measure examining ethnic exploration, ethnic belonging, cultural knowledge, and ethnic attachment using a Likert-type scale ranging from 1 ("Strongly disagree") to 7 ("Strongly agree"). The two subscales that indicated strong identification with Black culture—internalization Afrocentricity and

internalization multiculturalist exclusive—were used in the analyses. The CRIS has demonstrated excellent reliability (Cronbach's $\alpha$s = 0.65–0.88), construct validity (Cronbach's $\alpha$s = 0.71–0.87), and convergent validity (Cronbach's $\alpha$s = 0.72–0.89) among Black adolescents and young adults [50,51]. In the current sample, internalization Afrocentricity (Cronbach's $\alpha$ = 0.78) and internalization multiculturalist exclusive (Cronbach's $\alpha$ = 0.75) demonstrated excellent reliability.

## Statistical analyses

Analyses of variance and simple and multiple linear regressions were used. The independent variables were marketing (food, neutral), weight status (obese [BMI $\geq$ 95th percentile], non-overweight [BMI = 5th-84th percentile]), food-related ABs (highpal-nonfood, highpal-lowpal), or ethnic identity (internalization Afrocentricity, internalization multiculturalist exclusive). The dependent variable was energy consumption (kcal). Direct and interactional effects were examined. Participant characteristics included age and estimated household income. Data were screened for normality and extreme outliers recoded [52]. AB task trials with responses <200 ms, >2000 ms, or that were incorrect were excluded, as well as participants with error rates $\geq$10% [53]. Median bias scores (score = reaction time to least palatable image–reaction time to most palatable image) were calculated for highpal-nonfood and highpal-lowpal pairs. The AB task was coded using E-prime 2.0. Age, estimated household income, pre-visit self-reported intake, meal time, and pre-meal hunger were considered as covariates, but were not statistically associated with the outcome variables and so were not included in the models. Analyses were conducted using SPSS, Versions 24 and 25. Differences were considered significant at $p \leq 0.05$. All tests were two-tailed. Data, analytic methods, and study materials are available upon request by emailing the corresponding author.

## Results and discussion

### Participant characteristics

The total sample included 41 Black females 12–17 years (M, SD = 14.93, 1.67; Table 1). Based on parents' occupations, the median household income was estimated to be $96,110. BMI $z$-scores ranged from -1.01 to 2.62 (M, SD = 0.95, 1.00). There were no significant differences between groups.

### Aim 1: Assess feasibility of recruitment and protocol

We recruited a total of 167 adolescents primarily through direct mailings and pre-screened 89% of the potential participants (Fig 1). After excluding 56 (50.4%) of the potential participants, the remaining 55 (49.5%) adolescents were scheduled for a visit. Most ($n$ = 48, 87.3%)

**Table 1. Participants characteristics based on condition.**

| | n | Food Marketing (M, SD) | Neutral Marketing (M, SD) | p |
|---|---|---|---|---|
| **Age (y)** | 41 | 14.65 (1.78) | 15.28 (1.49) | .24 |
| **Estimated Household Income ($)** | 32 | 102595.56 (55526.65) | 94585.71 (31407.18) | .63 |
| **BMI$z$** | 41 | 1.00 (1.12) | 0.88 (0.84) | .71 |

M, mean; SD, standard deviation; y, years; $, dollars; BMI$z$, body mass index $z$-score.

*$p$ values significant at $p$ < .05.

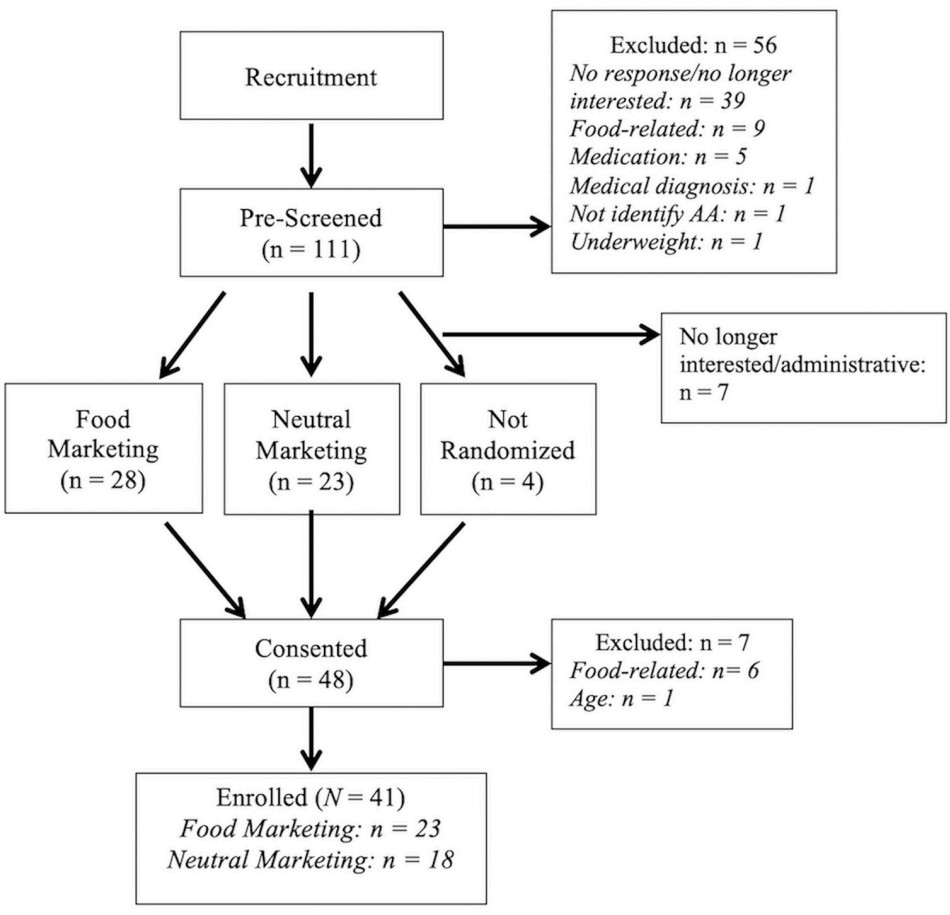

**Fig 1. Study flow.** Illustration detailing the flow of study participants from recruitment to participation.

attended the visit and signed consents/assents. The research team scheduled three touch points between the initial contact and study visit. These interactions provided continuous engagement with potential participants and the opportunity to reschedule visits, if needed. One hundred percent of adolescents reported following instructions to fast and not exercise two hours before the visit, and 100% of eligible participants completed the entire visit. Although all participants completed the AB task, seven (22.6%) participants were excluded from analyses due to high error rates.

## Aim 2: Examine initial effects of racially-targeted ads on energy intake

Across both conditions, participants consumed an average of 1083.65 kcal (SD = 351.84 kcal) during the test meal (Table 2), approximately 41.19% fat, 58.06 g of sugar, and 1444.52 mg of salt. Average pre-visit self-reported intake was 363.40 kcal (SD = 258.40 kcal). Thirty-three participants were included in the analyses for Aim 2a to examine the initial effects of exposure to racially-targeted ads on energy intake. We excluded eight participants ($n$ = 6 due to unsuccessful deception, $n$ = 2 due to missing meal duration data or meal duration greater than 30 minutes, and $n$ = 2 due to overweight status; two participants were excluded for more than one reason). Exposure to racially-targeted food ads (versus neutral ads) did not affect energy intake ($p$ > .99). *Post-hoc* analyses examined interactions based on weight. Although not statistically

**Table 2. Meal consumption.**

|  | n | Minimum | Maximum | Mean | Standard Deviation |
|---|---|---|---|---|---|
| **Total Calories (kcal)** | 33 | 477.30 | 1730.60 | 1083.65 | 351.84 |
| **Fat (%)** | 33 | 30.20 | 53.74 | 41.19 | 5.58 |
| **Sugar (g)** | 33 | 11.27 | 96.85 | 58.06 | 20.39 |
| **Salt (mg)** | 33 | 939.20 | 1932.72 | 1444.52 | 268.71 |
| **Meal duration (min)** | 33 | 8 | 27 | 17.00 | 5.60 |
| **Pre-Visit Intake** | 33 | 100.00 | 870.00 | 363.40 | 258.40 |

significant, adolescents with obesity consumed 239.75 kcal more than non-overweight adolescents ($F[1,29] = 2.92$, $p = 0.10$, partial $\eta^2 = 0.09$). Despite a non-significant interaction ($F[1,29] = 2.00$, $p = 0.17$, partial $\eta^2 = 0.06$; Fig 2), *among adolescents exposed to racially-targeted food ads*, adolescents with obesity consumed 399.58 kcal more than non-overweight adolescents. On the other hand, *among adolescents exposed to neutral ads*, adolescents with obesity consumed 37.94 kcal more than non-overweight adolescents.

## Aim 3: Examine initial effects of food-related abs on energy intake

Twenty-four participants were included in analyses for Aim 2b to examine initial effects of food-related ABs on energy intake. We excluded 17 participants ($n = 4$ due to missing ABs data, and $n = 7$ due to AB task error rates at or above 10%, in addition to the same eight participants excluded from Aim 2 analyses; three participants were excluded for more than one

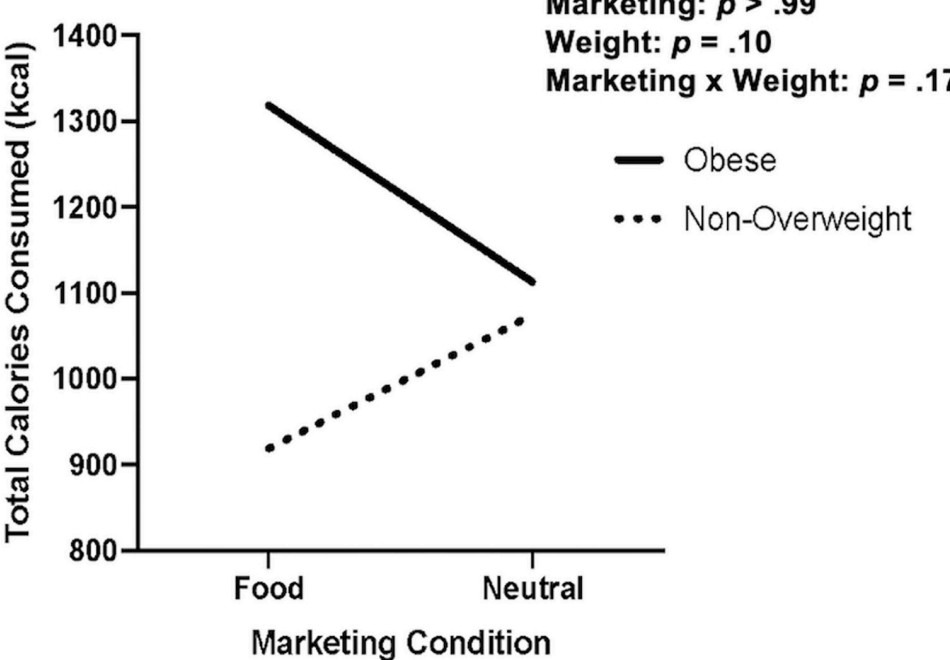

**Fig 2. Difference in consumption of calories among adolescents with and without obesity.** Difference in consumption of Calories among adolescents with and without obesity in the food and neutral marketing conditions. Among adolescents in the food marketing condition, adolescents with obesity consumed 399.58 kcal more than non-overweight adolescents compared to adolescents with obesity in the neutral marketing condition who consumed 37.94 kcal more than non-overweight adolescents. Although results were not significant, there was a small-to-medium effect size (partial $\eta^2 = .06$) for this difference.

reason). There were no significant initial effects of food-related ABs (highpal-nonfood bias, $p = 0.22$; highpal-lowfood bias, $p = 0.47$) or food-related ABs by racially-targeted food ads (highpal-nonfood bias, $p = 0.46$; highpal-lowpal bias, $p = 0.79$) on energy intake. *Post-hoc* analyses examined interactions based on weight. There was not a significant food-related ABs by obesity interaction (highpal-nonfood bias, $p = 0.40$; highpal-lowpal bias, $p = 0.25$) on energy intake.

### *Post-hoc* analyses examining interactions of ethnic identity

A total of 33 participants were included in *post-hoc* analyses to examine the interaction between marketing condition (racially-targeted food ad, neutral ad) and ethnic identity (internalization afrocentricity; internalization multiculturalist exclusive) on total energy intake. We excluded 8 participants (same as Aim 2). There was not a significant marketing condition by ethnic identity interaction on energy intake (internalization Afrocentricity, $p = 0.39$; internalization multiculturalist exclusive, $p = 0.64$).

## Discussion

Black youth and communities are inundated with food ads primarily promoting foods high in fat, sugar, and salt [54]. Many of these ads utilize culturally congruent messages that may be attractive to Black individuals [3,9]. Researchers have proposed a number of psychological theories that may elucidate the relationship between food commercials and behaviors. One theory that has not been fully explored among Black youth is food-related ABs [34]. Racially-targeted food marketing and other food cues that are pervasive in predominantly Black communities may place Black youth at risk of developing food-related ABs, which has been associated with weight and eating behaviors. The current project was conducted to examine the effect of racially-targeted food marketing and food-related ABs on eating behaviors among Black adolescent girls.

We found support for our first hypothesis that the study would be feasible based on high retention and completion rates of procedures. A larger study will require adjustments to the protocol to ensure alignment with current cultural trends and popular media. For instance, youth viewed an episode of the *Big Bang Theory*, a popular show among adolescents at the time of the study that included a predominantly White cast. Other TV shows that resonate with Black participants will likely be more suitable for the larger study.

We did not find support for our second aim regarding a significant effect of racially-targeted food ads on energy intake. We conducted a *post-hoc* analysis, however, where we found a trend for Black adolescent females with obesity consuming more energy when exposed to racially-targeted food ads compared to Black adolescent females with obesity who are exposed to neutral ads. There were small-to-medium effect sizes observed, and given the small sample size, it was difficult to reach significance. While psychological research typically makes efforts to find ways to increase effect size to improve probability of finding an effect (if one exists), smaller effect sizes may still have practical significance. It is common in biomedical clinical trials investigating serious diseases (e.g., heart attacks) to conclude that certain treatments are effective, even if the effect sizes are small [55]. The perspective of these researchers is that the benefit of even a small reduction in the risk of a heart attack significantly outweighs the risk of not being treated. While no study has examined food marketing research using this lens to date, it may be helpful to take a similar perspective. Even a small increase in the amount of food consumed after exposure to a food ad may underscore a risk that may outweigh the benefit of not making any changes to the current food environment. Even with a small effect size, if risks outweigh benefits, it is important for changes to be considered. It can also be argued that a small change detected during one meal, compounded across all meals over the course of an

entire lifetime is substantial. If such an impact is again compounded across millions of people, the economic impact may be quite large. More research is needed to quantify the population impact of food marketing research with smaller effect sizes. Black adolescents remain key targets of food and beverage marketing, despite nationwide efforts to address obesity and eating issues [1]. The current findings can provide information that can support future research in this area. Data from a larger study would be required to support this interpretation.

Our third aim was to examine food-related ABs as a potential mechanism by which food ads impact eating behaviors. We did not find a significant initial effect of high food-related ABs on energy intake or an interaction effect between exposure to racially-targeted ads and demonstrating high food-related ABs. Studies examining the association between food-related ABs and eating are mixed [56]. The significant AB and eating relationship has largely been conducted by examining intake after attention has been *manipulated* [56]. There may be a unique aspect of AB such that its effects on eating are more robustly captured when it is being changed in some way as opposed to examining its presence or absence. Capturing ABs in the laboratory can be challenging, which may also explain the inconsistencies in the literature [17,57] and non-significant results. Given the high error rate of the AB task, it will be important for future work to consider additional methods (e.g., neuroimaging) that can assess food-related ABs [23,57,58].

Limitations include the small sample size, which reduced our ability to detect statistically and clinically meaningful differences. Further, with small sample sizes, subgroup analyses can be misleading, and thus we have interpreted these data with caution. To prevent an effect of extreme hunger on the AB task [29], participants did not fast prior to the test meal, which made it difficult to adequately control for prior intake like other studies [44]. Although prior dietary intake was measured, there are several limitations to using such measures [59]. In the future, multiple visits may be beneficial to accurately examine energy intake after an overnight fast and ABs after satiation. Study strengths include a Black adolescent female sample, an understudied group who may be uniquely vulnerable to the effects of food ads. The study also included objective measurements of weight, food-related ABs, and consumption. It is important to continue work that acknowledges the multiple components that may influence obesity.

## Conclusions

The development of obesity linked to unhealthy eating behaviors [60] underscores the importance of sufficiently capturing factors that contribute to poor eating habits among Black adolescent females. Racially-targeted food ads and environmental food cues that are pervasive in predominantly Black communities may place these youth at risk of developing food-related ABs [37], which has been associated with weight and eating behaviors [20,21]. These data provide feasibility and initial findings of a pilot study investigating the impact of racially-targeted food ads and food-related ABs on eating behaviors in Black adolescent females.

The worsening rates of chronic diet-related illnesses calls for "strategic science" that has the capacity to fill knowledge gaps that can inform policies and have the potential to improve food environment [61]. Although the food industry has made varying commitments to improving nutritional quality of foods promoted on TV and other platforms, the majority of foods promoted still fail to meet nutritional guidelines for healthful living [62]. Research that helps uncover key questions and mechanisms that can identify levers of change may greatly assist capacity building for regulations or other means of promoting health in the current environment. Ultimately, these data have the potential to build scientific support towards important regulatory policies to encourage alignment of marketing with nutritional guidelines and limits on food marketing towards youth and racial/ethnic minorities.

## Supporting information

**S1 File. Minimal dataset.**
(SAV)

## Acknowledgments

We would like to acknowledge Dr. Cara Olsen for providing statistical consultation and the many families who participated in the research study.

## Author Contributions

**Conceptualization:** Omni Cassidy, Marian Tanofsky-Kraff, Andrew J. Waters, Marie Bragg, Tracy Sbrocco.

**Data curation:** Omni Cassidy, Andrew J. Waters, Lisa M. Shank.

**Formal analysis:** Omni Cassidy, Andrew J. Waters.

**Funding acquisition:** Omni Cassidy.

**Investigation:** Omni Cassidy, Lisa M. Shank, Abigail Pine, Mary Quattlebaum.

**Methodology:** Omni Cassidy, Marian Tanofsky-Kraff, Andrew J. Waters, Patrick H. DeLeon, Marie Bragg.

**Project administration:** Omni Cassidy.

**Resources:** Marian Tanofsky-Kraff, Marie Bragg, Tracy Sbrocco.

**Supervision:** Marian Tanofsky-Kraff, Andrew J. Waters, Patrick H. DeLeon, Tracy Sbrocco.

**Validation:** Marian Tanofsky-Kraff, Andrew J. Waters.

**Visualization:** Andrew J. Waters.

**Writing – original draft:** Omni Cassidy.

**Writing – review & editing:** Marian Tanofsky-Kraff, Andrew J. Waters, Lisa M. Shank, Abigail Pine, Mary Quattlebaum, Patrick H. DeLeon, Marie Bragg, Tracy Sbrocco.

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
