## [Decision Letter · Decision Letter 0]

6 Jan 2022

PONE-D-21-07262The impact of racially-targeted food marketing and attentional biases on consumption in Black adolescent females with and without obesity: Pilot data from the Black Adolescent & Entertainment (BAE) StudyPLOS ONE

Dear Dr. Cassidy,

Thank you for submitting your manuscript to PLOS ONE. After careful consideration, we feel that it has merit but does not fully meet PLOS ONE’s publication criteria as it currently stands. Therefore, we invite you to submit a revised version of the manuscript that addresses the points raised during the review process. Please submit your revised manuscript by Feb 20 2022 11:59PM. If you will need more time than this to complete your revisions, please reply to this message or contact the journal office at plosone@plos.org. Please include the following items when submitting your revised manuscript:A rebuttal letter that responds to each point raised by the academic editor and reviewer(s). You should upload this letter as a separate file labeled 'Response to Reviewers'.A marked-up copy of your manuscript that highlights changes made to the original version. You should upload this as a separate file labeled 'Revised Manuscript with Track Changes'.An unmarked version of your revised paper without tracked changes. You should upload this as a separate file labeled 'Manuscript'.

We look forward to receiving your revised manuscript.

Kind regards,

Ali B. Mahmoud, Ph.D.

Academic Editor

PLOS ONE

Journal Requirements:

"This research was funded by the National Institute of Minority Health and Health Disparities, grant number 1F31MD010675-01 (https://www.nimhd.nih.gov/); and Uniformed Services University of the Health Sciences, grant number TO7238251 (https://www.usuhs.edu/; both to O.C.). The funders had no role in study design, data collection and analysis, decision to publish, or preparation of the manuscript."

We note that one or more of the authors is affiliated with the funding organization, indicating the funder may have had some role in the design, data collection, analysis or preparation of your manuscript for publication; in other words, the funder played an indirect role through the participation of the co-authors. If the funding organization did not play a role in the study design, data collection and analysis, decision to publish, or preparation of the manuscript and only provided financial support in the form of authors' salaries and/or research materials, please do the following:

a. Review your statements relating to the author contributions, and ensure you have specifically and accurately indicated the role(s) that these authors had in your study. These amendments should be made in the online form.

b. Confirm in your cover letter that you agree with the following statement, and we will change the online submission form on your behalf: 

“The funder provided support in the form of salaries for authors [insert relevant initials], but did not have any additional role in the study design, data collection and analysis, decision to publish, or preparation of the manuscript. The specific roles of these authors are articulated in the ‘author contributions’ section.

Reviewers' comments:

Reviewer's Responses to Questions

**Comments to the Author**

1. Is the manuscript technically sound, and do the data support the conclusions?

Reviewer #1: Partly

Reviewer #2: Partly

2. Has the statistical analysis been performed appropriately and rigorously? 

Reviewer #1: I Don't Know

Reviewer #2: Yes

3. Have the authors made all data underlying the findings in their manuscript fully available?

Reviewer #1: Yes

Reviewer #2: Yes

4. Is the manuscript presented in an intelligible fashion and written in standard English?

Reviewer #1: Yes

Reviewer #2: Yes

5. Review Comments to the Author

Reviewer #1: This article shows a research aimed at assessing the feasibility of a pilot study testing the impact of racially-targeted food ads and food-related Attentional Biases on eating behaviors among a sample of black teenage females. Although the approach is very interesting, there are some aspects which are, according to my point of view, not clear and should need to be clarified/completed:

Introduction:

Although the first paragraph mentions the racial influence in food related habits, I would like to see more content on the social and economic influence of food habits. On the other hand, I would devote more space in the introduction to food related attentional biases and to the researches on this topic.

Methodology:

Is AB task a validated methodology? If it is, the reference must be included and if it is not, its origin must be explained.

The methodology section mentions a food preference questionnaire. I guess that it was used to exclude those individuals who did not like most of the food offered in the trial; however, the aim of this questionnaire is not clear enough.

Under which criteria were the racially-targeted food ads selected? It is not clear.

I think that it would be useful to include a figure with the methodological design including the different steps which have been taken and the different used tools at each step.

Although the methodology section includes the profession of the parents as a social and economical approach, this is not mentioned in the statistical part, why?

Results

Average self-reported daily intake: 340 kcal. I do not understand this data, which intake is being registered? A whole day intake?

On page 14 of the results, the following sentence is included … “A total of 33 participants were included in analyses to examine the impact of marketing condition and ethnic identity on total energy”

I find the assessment of the ethnic identity clear, but how was the impact of marketing condition assessed?

Conclusions. I think they need to be clearer and shorter. “Once participants were scheduled for visits, retention was excellent with nearly 90% attending the visit and signing consents/assents and 100% of participants completing the visit. The research team scheduled three touch points between the initial contact and study visit. These…..”. I would move this part to the Results section.

Minor issues:

Abstract: No acronyms must be included in the abstract (Attentional biases instead of AB)

References: Please, check Wootam et al. (DOI is too large)

Reviewer #2: The article addresses a topical issues. Food and dietary issues among ethnic communities in the USA are significant.

The authors build on an interesting data set which is reasonably well analysed. However, the paper has several areas for improvement before it can be accepted for publication.

0. I have not seen a dedicated introduction to the article. The authors need to start here in order to set the context and formulate their research questions.

1. The authors need to identify clear research questions and formulate hypotheses that are developed using the literature in support. Without these, it is difficult to put the results into context.

2. The literature review is ,limited in scope and breadth. While the authors have used a number of references linked to Black and Adolescent Entertainment (BAE) pilot study, a wider view of the literature is required. The authors could look at the literature on obesity in the US generally and as examining the black community in particular. They also need a critical approach. Some references could be made to the pandemic situation to see whether the hardship caused has compounded the situation.

3. The methodology is relatively well explained. However, as the participants were largely under 18 years old, it is important that the authors address the ethical issues and the various clearances they received prior to conducting the study.

4. Discussion: you need to pull together the main themes emerging from the analysis in a short discussion section before you move to the conclusion. This will show how your findings relate to existing literature and how your hypotheses have been supported or not.

Overall, you have some good data but the written paper needs strengthening. There are also a number of minor typos that require some attention.

6. PLOS authors have the option to publish the peer review history of their article (what does this mean?). If published, this will include your full peer review and any attached files.

Reviewer #1: No

Reviewer #2: **Yes: **Dieu Hack-Polay

---

## [Author Response · Author response to Decision Letter 0]

30 Jun 2022

Editor’s Comments:

1. Please ensure that your manuscript meets PLOS ONE’s style requirements, including those for file naming.

Done.

2a. Review your statements relating to the author contributions, and ensure you have specifically and accurately indicated the role(s) that these authors had in your study. These amendments should be made in the online form.

Done.

2b. Confirm in your cover letter that you agree with the following statement [funding statement], and we will change the online submission form on your behalf.

Done. 

3. In your Data Availability statement, you have not specified where the minimal data set underlying the results described in your manuscript can be found.

We have uploaded the minimal underlying data set as Supporting Information files. 

Reviewers’ Comments:

Reviewer #1: This article shows a research aimed at assessing the feasibility of a pilot study testing the impact of racially-targeted food ads and food-related Attentional Biases on eating behaviors among a sample of black teenage females. Although the approach is very interesting, there are some aspects which are, according to my point of view, not clear and should need to be clarified/completed:

Introduction:

1. Although the first paragraph mentions the racial influence in food related habits, I would like to see more content on the social and economic influence of food habits. On the other hand, I would devote more space in the introduction to food related attentional biases and to the researches on this topic. 

The introduction has been substantially reworked to provide a more thorough review and provide the foundation for the research questions and hypotheses. Please see below (lines 76-143):

Unhealthy food advertisements (“ads”) are one environmental factor that contributes to excessive weight gain and poor diet in Black youth.[1] Food companies spend nearly $11 billion each year on television (TV) food/beverage ads and $333 million on ads aimed at Black youth [1]. As such, Black youth are exposed to 86-119% more food/beverage TV ads than their White peers [1], and the majority of these ads are for products high in fat, sugar, and salt [2]. The highest promoters of food/beverages to Black youth include Yum! Brands (Taco Bell, KFC, Pizza Hut), 3G Capital (Burger King), McDonald’s, and Popeye’s [3, 4]. Ads placed in predominantly Black neighborhoods or feature Black actors are also less likely to promote healthy products compared to ads targeting non-Black audiences [5]. Increased exposure to unhealthy food and beverage ads influences attitudes, preferences, and consumption [6], contributing to poor diet and subsequently high rates of chronic illnesses in Black communities [7], It is critical to understand the potential influence of these ads on eating behaviors to prevent poor diet and determine appropriate intervention targets. 

Food ads targeting Black youth (hereafter referred to as “racially-targeted food ads”) are often embedded with cultural features intended to appeal to Black individuals. These ads may feature Black actors (e.g., spokespersons for Popeyes), celebrities (e.g., Beyoncé, LeBron James), music, or other activities (e.g., basketball) perceived to align with Black cultural preferences or values [5, 8-11]. Ads may, for example, feature young Black people who appear healthy, happy, and engaged in fun social activities (e.g., recording a music album) while eating fried chicken, hamburgers, or other fast foods. These “consumption symbols” are designed to highlight the psychosocial benefit of products—signaling to young Black people that consuming the advertised product will improve social connection, mood, or attractiveness [12, 13]. These messages may be particularly salient for adolescents whose higher-order cognitive functioning is still developing, and Black adolescents who may utilize the media more often to understand ethnic membership and support ethnic identity development [14]. Although research has supported the effect of unhealthy foods ads in children, there is a paucity of research among adolescents [15]. Further, the underlying mechanisms potentially driving effects are not well understood. 

Food-related attentional bias (ABs) is one underlying mechanism potentially driving the effect of food ads on attitudes, preferences, and consumption. Food-related ABs refer to distinct, yet related processes of selectively attending or over-attending to food cues in ones’ environment [16]. Although data are mixed [17-19], food-related ABs has been associated with overconsumption [20, 21], higher weight status [22, 23], and weight gain [24]. According to the incentive-sensitization theory [25], repeated exposure to food cues can lead to an heightened reward response in vulnerable individuals. This exacerbated response may subsequently increase salience and establish strong cravings for the food. As the salience for the food item increases, seeking out and consuming the food to diminish the craving becomes critical, overriding a person’s homeostatic feeding drive. With increased food availability in many food environments, such drives may lead to overeating [26, 27]. While most individuals exhibit food-related ABs towards fwdood-related cues when hungry [28-30], individuals repeatedly exposed to food cues [25] and those with obesity [29] may be particularly vulnerable. Not only are Black youth exposed to more unhealthy food ads compared to other groups [31], they often reside in communities with a disproportionately high density of food cues [32, 33]. Among Black youth with obesity [34, 35], external food cues may become highly salient, leading to food-related ABs and overeating [34, 36, 37]. Black youth who are exposed to racially-targeted food ads and have food-related ABs may be particularly vulnerable to overconsumption, although this relationship has not been explored in Black adolescent females with and without obesity. 

Studies elucidating the potential link between food ads and food-related ABs on energy consumption have primarily examined adults and are based on self-report [21, 38]. One experimental study, however, by Folkvord and colleagues [37] examined the interaction of unhealthy food ads and food-related ABs in children (grades 2 through 4). Children were randomized to an advergame (i.e., online video game promoting particular products) embedded with either a food ad or a non-food ad. Following the advergame, children were invited to consume snack foods ad libitum. Using eye tracking to measure food-related ABs, scientists measured three components of visual AB. Children who played the advergame with food ads consumed significantly more snack foods compared to children in the control condition. Among children who played the advergame, gaze duration (one component food-related AB) was significantly associated with more snack intake, even after controlling for age, sex, and hunger. Data suggest that food-related ABs may moderate the effect of food ads on eating, but this association is not well understood. 

To address the research gaps, the primary objectives of this pilot study were: 1) to assess feasibility of the study protocol to assess the intersection of racially-targeted foods ads and food-related ABs and 2) examine initial effects of racially-targeted food ads and food-related ABs on energy intake. Our primary hypotheses were that: 1) the study would be feasible, 2) exposure to racially-targeted food ads would be associated with higher energy intake, 3) exhibiting high food-related ABs would be associated with higher energy intake, and 4) there would be an interaction such that exposure to racially-targeted food ads and high food-related ABs would lead to the highest energy intake.

Methodology:

2. Is AB task a validated methodology? If it is, the reference must be included and if it is not, its origin must be explained.

Yes, the AB task is a validated methodology to assess food cue incentive salience. This particular task has also been validated in overweight samples. The following has been added on lines 232-233: “To examine ABs, participants completed a visual dot probe task, a validated behavioral measure of food cue incentive salience in overweight samples [29, 45-47].” 

3. The methodology section mentions a food preference questionnaire. I guess that it was used to exclude those individuals who did not like most of the food offered in the trial; however, the aim of this questionnaire is not clear enough. 

We have updated the section to say the following (lines 223-231):

To minimize the confounding effect of food preference on consumption from the test meal, only participants who preferred the majority of the foods offered during the test meal were included in the study. To assess food preferences, test meal food items were embedded within a larger, general food preference questionnaire that has been used to identify food preferences in youth.[44] Participants rated foods on a visual analogue scale ranging from 1 (“I hate this food”) to 10 (“I love this food”). Participants who did not provide a rating of 6 or more for at least 60% of the food items were excluded.

4. Under which criteria were the racially-targeted food ads selected? It is not clear. 

We have provided a more detailed description of the ad selection on lines 170-195:

The episode was embedded with: 1) four 30-second racially-targeted food ads (McDonald’s, Burger King, Popeye’s, Kentucky Fried Chicken) with culturally-congruent features (e.g., Black actors) or 2) four 30-second neutral ads (Geico, Esurance, Ford, Honda) without culturally-congruent features. Ads were selected from Adscope, a database that provides access to ad content [41]. The Adscope database categorizes commercials using the following criteria: name, date aired, platform (e.g., TV, radio), attributes (demographic specifications), markets (specific geographic regions), language, and length. Eight separate searches for food and neutral commercials was conducted using the following search terms: McDonald’s, Burger King, KFC, Popeye’s, Ford, Honda, Esurance, and Geico. Searches also included the following search terms (specific category included in parentheses): three years prior to search year in 2016 (date aired), TV only (media), ethnic targeted/African American (attributes), Baltimore, MD/Washington, DC/Hagerstown (markets), English (language), and 30 seconds (length). Commercials were excluded if they did not advertise the preferred items (e.g., McDonald advertising only chicken products), promoted activities that were unlikely to appeal to adolescents (e.g., toys for young children), references to sports or celebrities, off-season (e.g., references to winter), or promoted a corporation (e.g., Bank of America). For neutral commercials, ads that referenced food, eating, or restaurants were excluded. A total of 15 McDonald’s, four Burger King, 10 Popeye’s, eight KFC, nine Geico, 10 Ford, eight Esurance, and seven Honda commercials remained. The remaining commercials were reviewed based upon the following criteria: number of people, estimated screen time for persons and products, perceived age range and gender of persons on screen, and emotional affect. Increased exposure to food products could influence eating behaviors, and episodes can activate different emotions, such as happiness or sadness. The estimated time the product was on screen and emotional affect were, therefore, determined to be the most critical criteria. Various combinations of four food and four neutral commercials were compared until the optimal matching on the aforementioned categories was achieved.

5. I think that it would be useful to include a figure with the methodological design including the different steps which have been taken and the different used tools at each step. 

Done. 

6. Although the methodology section includes the profession of the parents as a social and economical approach, this is not mentioned in the statistical part, why? 

We have now included “estimated household income” in the statistical analysis section on line 273. 

Results

7. Average self-reported daily intake: 340 kcal. I do not understand this data, which intake is being registered? A whole day intake? 

We appreciate the Reviewer for requesting more clarification. The self-report intake was based on what participants consumed since awakening and prior to the study visit, which we assessed as a potential covariate. Including this data was necessary because participants only fasted 2 hours prior to their visit rather than overnight. We did not request that participants complete an overnight fast because all visits would then be required to start at one morning time to allow participants to break their fasting within an appropriate time frame. This time limitation would have significantly reduced the number of participants who could participate per day. We have removed “daily” from the description and changed it to “pre-visit.” This language has been updated throughout:

“Participants completed a self-report dietary recall of the foods and drinks they consumed since awakening.” (lines 221-222)

“The AB task was coded using E-prime 2.0. Age, SES, pre-visit self-reported intake….” (line 273)

“Average pre-visit self-reported intake was 363.40 kcal (SD = 258.40 kcal).” (line 319)

8. On page 14 of the results, the following sentence is included … “A total of 33 participants were included in analyses to examine the impact of marketing condition and ethnic identity on total energy” I find the assessment of the ethnic identity clear, but how was the impact of marketing condition assessed? 

We have updated the Results to the following: “A total of 33 participants were included in analyses to examine the interaction between marketing condition (racially-targeted food ad, neutral ad) and ethnic identity (internalization afrocentricity; internalization multiculturalist exclusive) on total energy intake.” (lines 349-354)

Conclusions

9a. I think they need to be clearer and shorter.

The discussion and conclusion have been substantially edited to summarize findings and improve clarity more adequately (lines 358-418):

Discussion

Black youth and communities are inundated with food ads primarily promoting foods high in fat, sugar, and salt [53]. Many of these ads utilize culturally congruent messages that may be attractive to Black individuals [3, 9]. Researchers have proposed a number of psychological theories that may elucidate the relationship between food commercials and behaviors. One theory that has not been fully explored among Black youth is food-related ABs [34]. Racially-targeted food marketing and other food cues that are pervasive in predominantly Black communities may place Black youth at risk of developing food-related ABs, which has been associated with weight and eating behaviors. The current project was conducted to examine the effect of racially-targeted food marketing and food-related ABs on eating behaviors among Black adolescent girls. 

We found support for our first hypothesis that the study would be feasible based on high retention and completion rates of procedures. A larger study will require some adjustments to the protocol to ensure alignment with current cultural trends and popular media. For instance, youth viewed an episode of the Big Bang Theory, a popular show among adolescents. Additional data supporting the popularity of shows that may resonate with Black participants may be more suitable for the larger study. 

We did not find a significant initial effect of racially-targeted food ads on energy intake. We conducted a post-hoc analysis where we found a trend with a small effect size. This result suggested that Black adolescent females with obesity may consume more energy when exposed to racially-targeted food ads compared to Black adolescent females with obesity who are exposed to neutral ads. Data from a larger study would be required to support this interpretation. We also examined food-related ABs as a potential mechanism by which food ads impact eating behaviors. We did not find a significant initial effect of high food-related ABs on energy intake or an interaction effect between exposure to racially-targeted ads and demonstrating high food-related ABs. Studies examining the association between food-related ABs and eating are mixed [54]. The significant AB and eating relationship has largely been conducted by examining intake after attention has been manipulated [54]. There may be a unique aspect of AB such that its effects on eating are more robustly captured when it is being changed in some way as opposed to examining its presence or absence. Capturing ABs in the laboratory can be challenging, which may also explain the inconsistencies in the literature [17, 55] and non-significant results. Given the high error rate of the AB task, it will also be important to consider additional methods (e.g., neuroimaging) that can assess food-related ABs [23, 55, 56]. Limitations include the small sample size, which reduced our ability to detect statistically and clinically meaningful differences. To prevent an effect of extreme hunger on the AB task[29], participants did not fast prior to the test meal, which made it difficult to adequately control for prior intake like other studies [43]. Although prior dietary intake was measured, there are several limitations to using such measures [57]. In the future, multiple visits may be beneficial to accurately examine energy intake after an overnight fast and ABs after satiation. Study strengths include a Black adolescent female sample, an understudied group who may be uniquely vulnerable to the effects of food ads. The study also included objective measurements of weight, food-related ABs, and consumption. It is important to continue work that acknowledges the multiple components that may influence obesity. 

Conclusions

The development of obesity linked to unhealthy eating behaviors [58] underscores the importance of sufficiently capturing factors that contribute to poor eating habits among Black adolescent females. Racially-targeted food ads and environmental food cues that are pervasive in predominantly Black communities may place these youth at risk of developing food-related ABs [37], which has been associated with weight and eating behaviors [20, 21]. These data provide feasibility and initial findings of a pilot study investigating the impact of racially-targeted food ads and food-related ABs on eating behaviors in Black adolescent females.

The worsening rates of chronic diet-related illnesses calls for “strategic science” that has the capacity to fill knowledge gaps that can inform policies and have the potential to improve food environment [59]. Although the food industry has made varying commitments to improving nutritional quality of foods promoted on TV and other platforms, the majority of foods promoted still fail to meet nutritional guidelines for healthful living [60]. Research that helps uncover key questions and mechanisms that can identify levers of changes may greatly assist capacity building for regulations or other means of promoting health in the current environment. Ultimately, these data have the potential to build scientific support towards important regulatory policies to encourage alignment of marketing with nutritional guidelines and limits on food marketing towards youth and racial/ethnic minorities. 

9b. “Once participants were scheduled for visits, retention was excellent with nearly 90% attending the visit and signing consents/assents and 100% of participants completing the visit. The research team scheduled three touch points between the initial contact and study visit. These…..”. I would move this part to the Results section. 

The sentences have been removed from the Conclusion and placed in the Results section (lines 300-307).

Minor issues:

10. Abstract: No acronyms must be included in the abstract (Attentional biases instead of AB) 

Done.

11. References: Please, check Wootam et al. (DOI is too large) 

 Done.

Reviewer #2: The article addresses a topical issues. Food and dietary issues among ethnic communities in the USA are significant. The authors build on an interesting data set which is reasonably well analysed. However, the paper has several areas for improvement before it can be accepted for publication.

We thank the Reviewer for the comments.

0. I have not seen a dedicated introduction to the article. The authors need to start here in order to set the context and formulate their research questions. 

The introduction has been substantially reworked to provide a more thorough review and provide the foundation for the research questions and hypotheses. Please see below (lines 76-143):

Unhealthy food advertisements (“ads”) are one environmental factor that contributes to excessive weight gain and poor diet in Black youth.[1] Food companies spend nearly $11 billion each year on television (TV) food/beverage ads and $333 million on ads aimed at Black youth [1]. As such, Black youth are exposed to 86-119% more food/beverage TV ads than their White peers [1], and the majority of these ads are for products high in fat, sugar, and salt [2]. The highest promoters of food/beverages to Black youth include Yum! Brands (Taco Bell, KFC, Pizza Hut), 3G Capital (Burger King), McDonald’s, and Popeye’s [3, 4]. Ads placed in predominantly Black neighborhoods or feature Black actors are also less likely to promote healthy products compared to ads targeting non-Black audiences [5]. Increased exposure to unhealthy food and beverage ads influences attitudes, preferences, and consumption [6], contributing to poor diet and subsequently high rates of chronic illnesses in Black communities [7], It is critical to understand the potential influence of these ads on eating behaviors to prevent poor diet and determine appropriate intervention targets. 

Food ads targeting Black youth (hereafter referred to as “racially-targeted food ads”) are often embedded with cultural features intended to appeal to Black individuals. These ads may feature Black actors (e.g., spokespersons for Popeyes), celebrities (e.g., Beyoncé, LeBron James), music, or other activities (e.g., basketball) perceived to align with Black cultural preferences or values [5, 8-11]. Ads may, for example, feature young Black people who appear healthy, happy, and engaged in fun social activities (e.g., recording a music album) while eating fried chicken, hamburgers, or other fast foods. These “consumption symbols” are designed to highlight the psychosocial benefit of products—signaling to young Black people that consuming the advertised product will improve social connection, mood, or attractiveness [12, 13]. These messages may be particularly salient for adolescents whose higher-order cognitive functioning is still developing, and Black adolescents who may utilize the media more often to understand ethnic membership and support ethnic identity development [14]. Although research has supported the effect of unhealthy foods ads in children, there is a paucity of research among adolescents [15]. Further, the underlying mechanisms potentially driving effects are not well understood. 

Food-related attentional bias (ABs) is one underlying mechanism potentially driving the effect of food ads on attitudes, preferences, and consumption. Food-related ABs refer to distinct, yet related processes of selectively attending or over-attending to food cues in ones’ environment [16]. Although data are mixed [17-19], food-related ABs has been associated with overconsumption [20, 21], higher weight status [22, 23], and weight gain [24]. According to the incentive-sensitization theory [25], repeated exposure to food cues can lead to an heightened reward response in vulnerable individuals. This exacerbated response may subsequently increase salience and establish strong cravings for the food. As the salience for the food item increases, seeking out and consuming the food to diminish the craving becomes critical, overriding a person’s homeostatic feeding drive. With increased food availability in many food environments, such drives may lead to overeating [26, 27]. While most individuals exhibit food-related ABs towards fwdood-related cues when hungry [28-30], individuals repeatedly exposed to food cues [25] and those with obesity [29] may be particularly vulnerable. Not only are Black youth exposed to more unhealthy food ads compared to other groups [31], they often reside in communities with a disproportionately high density of food cues [32, 33]. Among Black youth with obesity [34, 35], external food cues may become highly salient, leading to food-related ABs and overeating [34, 36, 37]. Black youth who are exposed to racially-targeted food ads and have food-related ABs may be particularly vulnerable to overconsumption, although this relationship has not been explored in Black adolescent females with and without obesity. 

Studies elucidating the potential link between food ads and food-related ABs on energy consumption have primarily examined adults and are based on self-report [21, 38]. One experimental study, however, by Folkvord and colleagues [37] examined the interaction of unhealthy food ads and food-related ABs in children (grades 2 through 4). Children were randomized to an advergame (i.e., online video game promoting particular products) embedded with either a food ad or a non-food ad. Following the advergame, children were invited to consume snack foods ad libitum. Using eye tracking to measure food-related ABs, scientists measured three components of visual AB. Children who played the advergame with food ads consumed significantly more snack foods compared to children in the control condition. Among children who played the advergame, gaze duration (one component food-related AB) was significantly associated with more snack intake, even after controlling for age, sex, and hunger. Data suggest that food-related ABs may moderate the effect of food ads on eating, but this association is not well understood. 

To address the research gaps, the primary objectives of this pilot study were: 1) to assess feasibility of the study protocol to assess the intersection of racially-targeted foods ads and food-related ABs and 2) examine initial effects of racially-targeted food ads and food-related ABs on energy intake. Our primary hypotheses were that: 1) the study would be feasible, 2) exposure to racially-targeted food ads would be associated with higher energy intake, 3) exhibiting high food-related ABs would be associated with higher energy intake, and 4) there would be an interaction such that exposure to racially-targeted food ads and high food-related ABs would lead to the highest energy intake.

1. The authors need to identify clear research questions and formulate hypotheses that are developed using the literature in support. Without these, it is difficult to put the results into context. 

Please see comment #0 above.

2. The literature review is limited in scope and breadth. While the authors have used a number of references linked to Black and Adolescent Entertainment (BAE) pilot study, a wider view of the literature is required. The authors could look at the literature on obesity in the US generally and as examining the black community in particular. They also need a critical approach. Some references could be made to the pandemic situation to see whether the hardship caused has compounded the situation. 

Please see comment #0 above.

3. The methodology is relatively well explained. However, as the participants were largely under 18 years old, it is important that the authors address the ethical issues and the various clearances they received prior to conducting the study. 

We appreciate the Reviewer’s comment. We have updated the Method section with more details (lines 203-211): 

The study was approved by the Human Research Protection Program at the Uniformed Services University and underwent an ethical review to ensure proper protections for minors. Study team members verbally described the procedures, risks, and benefits of the study with parents/guardians and adolescents. A parent/guardian provided written informed consent and adolescents provided written informed assent. Study team members emphasized to parents/guardians and adolescents that participation was completely voluntary and they could withdraw from the study without penalty or effect. We emphasized that adolescents had the right to decline participation or withdraw from the study at any point regardless of the parents’/guardians’ interest in their participation. 

4. Discussion: you need to pull together the main themes emerging from the analysis in a short discussion section before you move to the conclusion. This will show how your findings relate to existing literature and how your hypotheses have been supported or not. 

The discussion and conclusion have been substantially edited to more adequately summarize findings and improve clarity (lines 358-418):

Discussion

Black youth and communities are inundated with food ads primarily promoting foods high in fat, sugar, and salt [53]. Many of these ads utilize culturally congruent messages that may be attractive to Black individuals [3, 9]. Researchers have proposed a number of psychological theories that may elucidate the relationship between food commercials and behaviors. One theory that has not been fully explored among Black youth is food-related ABs [34]. Racially-targeted food marketing and other food cues that are pervasive in predominantly Black communities may place Black youth at risk of developing food-related ABs, which has been associated with weight and eating behaviors. The current project was conducted to examine the effect of racially-targeted food marketing and food-related ABs on eating behaviors among Black adolescent girls. 

We found support for our first hypothesis that the study would be feasible based on high retention and completion rates of procedures. A larger study will require some adjustments to the protocol to ensure alignment with current cultural trends and popular media. For instance, youth viewed an episode of the Big Bang Theory, a popular show among adolescents. Additional data supporting the popularity of shows that may resonate with Black participants may be more suitable for the larger study. 

We did not find a significant initial effect of racially-targeted food ads on energy intake. We conducted a post-hoc analysis where we found a trend with a small effect size. This result suggested that Black adolescent females with obesity may consume more energy when exposed to racially-targeted food ads compared to Black adolescent females with obesity who are exposed to neutral ads. Data from a larger study would be required to support this interpretation. We also examined food-related ABs as a potential mechanism by which food ads impact eating behaviors. We did not find a significant initial effect of high food-related ABs on energy intake or an interaction effect between exposure to racially-targeted ads and demonstrating high food-related ABs. Studies examining the association between food-related ABs and eating are mixed [54]. The significant AB and eating relationship has largely been conducted by examining intake after attention has been manipulated [54]. There may be a unique aspect of AB such that its effects on eating are more robustly captured when it is being changed in some way as opposed to examining its presence or absence. Capturing ABs in the laboratory can be challenging, which may also explain the inconsistencies in the literature [17, 55] and non-significant results. Given the high error rate of the AB task, it will also be important to consider additional methods (e.g., neuroimaging) that can assess food-related ABs [23, 55, 56]. Limitations include the small sample size, which reduced our ability to detect statistically and clinically meaningful differences. To prevent an effect of extreme hunger on the AB task[29], participants did not fast prior to the test meal, which made it difficult to adequately control for prior intake like other studies [43]. Although prior dietary intake was measured, there are several limitations to using such measures [57]. In the future, multiple visits may be beneficial to accurately examine energy intake after an overnight fast and ABs after satiation. Study strengths include a Black adolescent female sample, an understudied group who may be uniquely vulnerable to the effects of food ads. The study also included objective measurements of weight, food-related ABs, and consumption. It is important to continue work that acknowledges the multiple components that may influence obesity. 

Conclusions

The development of obesity linked to unhealthy eating behaviors [58] underscores the importance of sufficiently capturing factors that contribute to poor eating habits among Black adolescent females. Racially-targeted food ads and environmental food cues that are pervasive in predominantly Black communities may place these youth at risk of developing food-related ABs [37], which has been associated with weight and eating behaviors [20, 21]. These data provide feasibility and initial findings of a pilot study investigating the impact of racially-targeted food ads and food-related ABs on eating behaviors in Black adolescent females.

The worsening rates of chronic diet-related illnesses calls for “strategic science” that has the capacity to fill knowledge gaps that can inform policies and have the potential to improve food environment [59]. Although the food industry has made varying commitments to improving nutritional quality of foods promoted on TV and other platforms, the majority of foods promoted still fail to meet nutritional guidelines for healthful living [60]. Research that helps uncover key questions and mechanisms that can identify levers of changes may greatly assist capacity building for regulations or other means of promoting health in the current environment. Ultimately, these data have the potential to build scientific support towards important regulatory policies to encourage alignment of marketing with nutritional guidelines and limits on food marketing towards youth and racial/ethnic minorities. 

5. Overall, you have some good data but the written paper needs strengthening. There are also a number of minor typos that require some attention. 

We appreciate the Reviewers’ comments. We believe the edits we have made have strengthened the manuscript. We have also reviewed the paper and updated the minor typos.

---

## [Decision Letter · Decision Letter 1]

27 Sep 2022

PONE-D-21-07262R1The impact of racially-targeted food marketing and attentional biases on consumption in Black adolescent females with and without obesity: Pilot data from the Black Adolescent & Entertainment (BAE) StudyPLOS ONE

Dear Dr. Cassidy,

Thank you for submitting your manuscript to PLOS ONE. After careful consideration, we feel that it has merit but does not fully meet PLOS ONE’s publication criteria as it currently stands. Therefore, we invite you to submit a revised version of the manuscript that addresses the points raised during the review process.

We look forward to receiving your revised manuscript.

Kind regards,

Ali B. Mahmoud, Ph.D.

Academic Editor

PLOS ONE

Journal Requirements:

Reviewers' comments:

Reviewer's Responses to Questions

**Comments to the Author**

1. If the authors have adequately addressed your comments raised in a previous round of review and you feel that this manuscript is now acceptable for publication, you may indicate that here to bypass the “Comments to the Author” section, enter your conflict of interest statement in the “Confidential to Editor” section, and submit your "Accept" recommendation.

Reviewer #2: All comments have been addressed

Reviewer #3: (No Response)

2. Is the manuscript technically sound, and do the data support the conclusions?

Reviewer #2: Yes

Reviewer #3: Yes

3. Has the statistical analysis been performed appropriately and rigorously? 

Reviewer #2: Yes

Reviewer #3: Yes

4. Have the authors made all data underlying the findings in their manuscript fully available?

Reviewer #2: Yes

Reviewer #3: (No Response)

5. Is the manuscript presented in an intelligible fashion and written in standard English?

Reviewer #2: Yes

Reviewer #3: Yes

6. Review Comments to the Author

Reviewer #2: Thank you to the authors for responding to the comments made. The response is thorough and strengthens the paper.

Reviewer #3: Please note: I am reading this paper for the first time after its first round of revisions. Apologies if any of my recommendations contradict previous reviewers' - which can happen!

Small note:

Unhealthy food advertisements (“ads”) suggested to me that “ads” is short for the 3-word phrase; suggest rewriting

Abstract

I found some aspects of the Abstract confusing and suggest it could be clarified by attending to sequencing of information

1.

‘Fifty-five (50%) adolescents qualified for the study.’

What enabled them to qualify and what were their characteristics vs those who did not? Is this information key to the study outcomes? The presence of 'qualified' suggests it is.

Earlier it states participants were N=31; later that 87% participated; and then that all eligible participants completed the visit. I can’t see how these figures are related to one another.

Having read on I see that the method states N=41, so perhaps above is a typo, and this explains 87%; but the other questions remain.

Furthermore, some of the above information is not given in the Method, which doesn’t help.

I suggest you remove most of this from the Abstract, giving only the correct participant number, and saving Abstract word count for more substantive matters.

And explain about recruitment and participation etc in the Method

2.

‘Despite a non-significant interaction, preliminary data suggested that adolescents with

obesity may be particularly vulnerable to racially-targeted foods ads (p=0.17)’

Again, step-wise sequencing of information would clarify. Suggest you lead with main effects and then interaction, or non-significant findings and then significant ones – at present it is difficult to absorb the findings quickly, which is what one needs from an Abstract.

3.

Finally, the concluding sentence of the Abstract is rather general. Something that indicates which data and which direction this is taking you would help the reader.

4.

Participant characteristics –

Was this itself an outcome? Or was the hypothesis about recruitment, retention, completion?

Either way it would make more sense to section out the Results according to hypotheses/research questions so the purpose of each section is clear.

5.

‘285 Based on parents’ occupations, the median income was $96,110’ etc

What is the source for this?

Furthermore, it’s difficult to gain a rapid overview of this information in the narrative format. I would consider placing all the relevant demographic information in a single table.

6.

‘304 …. One hundred percent of adolescents followed instructions to fast and

305 not exercise before the visit’

I assume it might be more accurate to say that they reported having followed the instructions?

7.

‘319.. Average pre-visit self-reported intake was 363.40 kcal (SD = 258.40 kcal).’

How does this align with instructions to fast above?

8.

336 onwards – this seems to repeat the findings just reported?

Overall this section is really not very clear, and I had to read it several times to grasp it. I suggest you consider a rewrite with clear subheads and topic sentence introducing purpose of each analysis.

9.

371 – Big Bang Theory – perhaps say a popular show among adolescents at the time but one with an all-White cast? Among the young people I know, BBT has become quite discredited not just for its lack of diversity but also for its stereotyping of gender and neurodiversity.

9.

Discussion – I think you could consider discussing the literature on media effect sizes and issues about how sub-group effects are washed out in all-group analyses. Overall, a stronger account of the meaning and implications of this complex study, linking it to current literature and pointing forward to future work, would strengthen the paper in my view.

10. Some Minor things:

83 ‘or feature Black actors’ should read ‘or featuring’

115 ‘towards fwdood’

7. PLOS authors have the option to publish the peer review history of their article (what does this mean?). If published, this will include your full peer review and any attached files.

Reviewer #2: **Yes: **Prof. Dieu Hack-Polay

Reviewer #3: No

---

## [Author Response · Author response to Decision Letter 1]

11 Nov 2022

November 11, 2022

Ali B. Mahmoud, Ph.D.

Academic Editor

PLOS ONE

Dear Dr. Mahmoud,

We appreciate another opportunity to revise and resubmit our manuscript entitled, “The impact of racially-targeted food marketing and attentional biases on consumption in Black adolescent females with and without obesity: Pilot data from the Black Adolescent & Entertainment (BAE) Study” (PONE-D-21-07262R1). 

We have outlined our responses to Reviewer #3’s comments. We have placed the comments in bold and our responses in normal font:

Review Comments to the Author

Reviewer #3: Please note: I am reading this paper for the first time after its first round of revisions. Apologies if any of my recommendations contradict previous reviewers' - which can happen!

Small note - Unhealthy food advertisements (“ads”) suggested to me that “ads” is short for the 3-word phrase; suggest rewriting

We have added, ‘ “advertisements” hereafter referred to as “ads” ’ to clarify the shortened phrase.

Abstract - I found some aspects of the Abstract confusing and suggest it could be clarified by attending to sequencing of information

1. ‘Fifty-five (50%) adolescents qualified for the study.’ What enabled them to qualify and what were their characteristics vs those who did not? Is this information key to the study outcomes? The presence of 'qualified' suggests it is. Earlier it states participants were N=31; later that 87% participated; and then that all eligible participants completed the visit. I can’t see how these figures are related to one another. Having read on I see that the method states N=41, so perhaps above is a typo, and this explains 87%; but the other questions remain. Furthermore, some of the above information is not given in the Method, which doesn’t help. I suggest you remove most of this from the Abstract, giving only the correct participant number, and saving Abstract word count for more substantive matters. And explain about recruitment and participation etc in the Method.

We have edited the sample size typo and removed most of these details from the Abstract, which now states (lines 53-73):

Unhealthy food advertisements (“advertisements” hereafter referred to as “ads”) are linked to poor diet and obesity, and food companies disproportionally target Black youth. Little is known about the mechanisms whereby food ads influence diet. One possibility may be racially-targeted ads that appeal to Black youth. Those with food-related attentional biases may be especially vulnerable. The objective of this project was to assess the feasibility and initial effects of a pilot study testing the influence of racially-targeted food ads and food-related attentional biases on eating behaviors among a sample of Black adolescent females. Feasibility of recruitment, retention, and procedures were examined. Participants (N=41, 12-17y) were randomized to view a television episode clip of the Big Bang Theory embedded with either four 30-second racially-targeted food ads or neutral ads. A computer dot probe task assessed food-related attentional biases. The primary outcome was caloric consumption from a laboratory test meal. Interactions based on weight and ethnic identity were also examined. Analyses of variance and regressions were used to assess main and interaction effects. Exposure to racially-targeted food ads (versus neutral ads) did not affect energy consumption (p > .99). Although not statistically significant, adolescents with obesity consumed nearly 240 kcal more than non-overweight adolescents (p = 0.10). There were no significant preliminary effects related to food-related attentional biases or ethnic identity (ps=0.22-0.79). Despite a non-significant interaction, these data provide preliminary support that adolescents with obesity may be particularly vulnerable to racially-targeted food ads. An adequately powered trial is necessary to further elucidate the associations among racially-targeted food ads among Black adolescent girls with obesity.

Information about the recruitment is included in the Results (versus the Method) per the suggestion of a previous reviewer. The paragraph states (lines 294-303):

We recruited a total of 167 adolescents primarily through direct mailings and pre-screened 89% of the potential participants (Fig 1). After excluding 56 (50.4%) of the potential participants, the remaining 55 (49.5%) adolescents were scheduled for a visit. Most (n = 48, 87.3%) attended the visit and signed consents/assents. The research team scheduled three touch points between the initial contact and study visit. These interactions provided continuous engagement with potential participants and the opportunity to reschedule visits, if needed. One hundred percent of adolescents reported following instructions to fast and not exercise two hours before the visit, and 100% of eligible participants completed the entire visit. Although all participants completed the AB task, seven (22.6%) participants were excluded from analyses due to high error rates.

2. ‘Despite a non-significant interaction, preliminary data suggested that adolescents with

obesity may be particularly vulnerable to racially-targeted foods ads (p=0.17)’ Again, step-wise sequencing of information would clarify. Suggest you lead with main effects and then interaction, or non-significant findings and then significant ones – at present it is difficult to absorb the findings quickly, which is what one needs from an Abstract.

We have updated the Abstract to provide a step-sequencing of information about the results (lines 64-69):

Analyses of variance and regressions were used to assess main and interaction effects. Exposure to racially-targeted food ads (versus neutral ads) did not affect energy consumption (p > .99). Although not statistically significant, adolescents with obesity consumed nearly 240 kcal more than non-overweight adolescents (p = 0.10). There were no significant preliminary effects related to food-related attentional biases or ethnic identity (ps=0.22-0.79).

3. Finally, the concluding sentence of the Abstract is rather general. Something that indicates which data and which direction this is taking you would help the reader.

We have edited the concluding sentence to the following (lines 69-73): 

Despite a non-significant interaction, these data provide preliminary support that adolescents with obesity may be particularly vulnerable to racially-targeted food ads. An adequately powered trial is necessary to further elucidate the associations among racially-targeted food ads among Black adolescent girls with obesity.

4. Participant characteristics –Was this itself an outcome? Or was the hypothesis about recruitment, retention, completion? Either way it would make more sense to section out the Results according to hypotheses/research questions so the purpose of each section is clear.

The participant characteristics section provides the relevant demographic data, but was not itself an outcome. The Results have been re-organized based on aims to make each section and the analyses clearer. See lines 281-353.

5. ‘285 Based on parents’ occupations, the median income was $96,110’ etc What is the source for this? Furthermore, it’s difficult to gain a rapid overview of this information in the narrative format. I would consider placing all the relevant demographic information in a single table.

The source of the estimated income is from the 2018 national wage estimates provided by the U.S. Bureau of Labor Statistics (citation added; lines 214-215). Upon further reflection, we recognized that percentage breakdown is not relevant for these analyses and so has been removed.

6.‘304 …. One hundred percent of adolescents followed instructions to fast and 305 not exercise before the visit’ I assume it might be more accurate to say that they reported having followed the instructions?

The sentence now states (lines 299-301), “One hundred percent of adolescents reported following instructions to fast and not exercise…”

7. ‘319... Average pre-visit self-reported intake was 363.40 kcal (SD = 258.40 kcal).’ How does this align with instructions to fast above?

We asked participants not to fast two hours before their visit, so some participants may have eaten prior to the 2-hour window. This detail has been added for clarity (lines 299-301): “One hundred percent of adolescents reported following instructions to fast and not exercise two hours before the visit…”

8. 336 onwards – this seems to repeat the findings just reported? Overall this section is really not very clear, and I had to read it several times to grasp it. I suggest you consider a rewrite with clear subheads and topic sentence introducing purpose of each analysis.

We have reorganized these sections and created subheadings for clarity. See lines See lines 292-353.

9. 371 – Big Bang Theory – perhaps say a popular show among adolescents at the time but one with an all-White cast? Among the young people I know, BBT has become quite discredited not just for its lack of diversity but also for its stereotyping of gender and neurodiversity.

We now state (lines 368-371):

For instance, youth viewed an episode of the Big Bang Theory, a popular show among adolescents at the time of the study that included a predominantly White cast. Other TV shows that resonate with Black participants will likely be more suitable for the larger study. 

9. Discussion – I think you could consider discussing the literature on media effect sizes and issues about how sub-group effects are washed out in all-group analyses. Overall, a stronger account of the meaning and implications of this complex study, linking it to current literature and pointing forward to future work, would strengthen the paper in my view.

We have added a more thorough discussion of effect sizes (lines 376-394):

There were small-to-medium effect sizes observed, and given the small sample size, it was difficult to reach significance. While psychological research typically makes efforts to find ways to increase effect size to improve probability of finding an effect (if one exists), smaller effect sizes may still have practical significance. It is common in biomedical clinical trials investigating serious diseases (e.g., heart attacks) to conclude that certain treatments are effective, even if the effect sizes are small [55]. The perspective of these researchers is that the benefit of even a small reduction in the risk of a heart attack significantly outweighs the risk of not being treated. While no study has examined food marketing research using this lens to date, it may be helpful to take a similar perspective. Even a small increase in the amount of food consumed after exposure to a food ad may underscore a risk that may outweigh the benefit of not making any changes to the current food environment. Even with a small effect size, if risks outweigh benefits, it is important for changes to be considered. It can also be argued that a small change detected during one meal, compounded across all meals over the course of an entire lifetime is substantial. If such an impact is again compounded across millions of people, the economic impact may be quite large. More research is needed to quantify the population impact of food marketing research with smaller effect sizes. Black adolescents remain key targets of food and beverage marketing, despite nationwide efforts to address obesity and eating issues [1]. The current findings can provide information that can support future research in this area.

We added a comment about the limitations of sub-group analyses to the Limitations (lines 408-409), “Further, with small sample sizes, subgroup analyses can be misleading, and thus we have interpreted these data with caution.”

Given the small sample size and focus on preliminary examinations, prior reviewers cautioned against making any additional interpretations and implications from these data. 

10. Some Minor things: 83 ‘or feature Black actors’ should read ‘or featuring’ 115 ‘towards fwdood’

Corrected.

We appreciate the Reviewer’s comments and hope our updates have strengthened the manuscript.

Best,

Omni Cassidy, Ph.D.

---

## [Decision Letter · Decision Letter 2]

19 Dec 2022

The impact of racially-targeted food marketing and attentional biases on consumption in Black adolescent females with and without obesity: Pilot data from the Black Adolescent & Entertainment (BAE) Study

PONE-D-21-07262R2

Dear Dr. Cassidy,

We’re pleased to inform you that your manuscript has been judged scientifically suitable for publication and will be formally accepted for publication once it meets all outstanding technical requirements.

Kind regards,

Ali B. Mahmoud, Ph.D.

Academic Editor

PLOS ONE

Additional Editor Comments (optional):

Reviewers' comments:

Reviewer's Responses to Questions

**Comments to the Author**

1. If the authors have adequately addressed your comments raised in a previous round of review and you feel that this manuscript is now acceptable for publication, you may indicate that here to bypass the “Comments to the Author” section, enter your conflict of interest statement in the “Confidential to Editor” section, and submit your "Accept" recommendation.

Reviewer #3: All comments have been addressed

2. Is the manuscript technically sound, and do the data support the conclusions?

Reviewer #3: (No Response)

3. Has the statistical analysis been performed appropriately and rigorously? 

Reviewer #3: (No Response)

4. Have the authors made all data underlying the findings in their manuscript fully available?

Reviewer #3: (No Response)

5. Is the manuscript presented in an intelligible fashion and written in standard English?

Reviewer #3: (No Response)

6. Review Comments to the Author

Reviewer #3: (No Response)

7. PLOS authors have the option to publish the peer review history of their article (what does this mean?). If published, this will include your full peer review and any attached files.

Reviewer #3: No

---

## [Editor Report · Acceptance letter]

11 Jan 2023

PONE-D-21-07262R2 

The impact of racially-targeted food marketing and attentional biases on consumption in Black adolescent females with and without obesity:  Pilot data from the Black Adolescent & Entertainment (BAE) Study 

Dear Dr. Cassidy:

I'm pleased to inform you that your manuscript has been deemed suitable for publication in PLOS ONE. Congratulations! Your manuscript is now with our production department. 

Kind regards, 

on behalf of

Dr. Ali B. Mahmoud 

Academic Editor

PLOS ONE